Impact of bacterial culture medium on composition and characteristics of Burkholderia pseudomallei extracellular polymeric substances

Srithabut Suthantip 1
Chaianunporn Thotsapol 1
Chareonsudjai Sorujsiri 2
Chareonsudjai Pisit pisit@kku.ac.th 1
1 Department of Environmental Science, Faculty of Science, Khon Kaen University , Khon Kaen , Thailand
2 Department of Microbiology, Faculty of Medicine, Khon Kaen University , Khon Kaen , Thailand
Flores-Valdez Mario Alberto
Electronic publication date: 2025 Dec 16
Publication date: 2025
Volume: 13
Electronic Location ID: e20488
Received 2025 Aug 27; Accepted 2025 Nov 7
Copyright: ©2025 Srithabut et al.
Copyright year: 2025
Copyright holder: Srithabut et al.
License: This is an open access article distributed under the terms of the Creative Commons Attribution License, which permits unrestricted use, distribution, reproduction and adaptation in any medium and for any purpose provided that it is properly attributed. For attribution, the original author(s), title, publication source (PeerJ) and either DOI or URL of the article must be cited.
License URL: https://creativecommons.org/licenses/by/4.0/

Keywords: Burkholderia pseudomallei, Culture medium, Extracellular polymeric substances, Biofilm, Carbon to nitrogen ratio

Funding: Khon Kaen University Research Grant No. 6200021002 This work was supported by grants from the Khon Kaen University Research Grant (Grant No. 6200021002). The funders had no role in study design, data collection and analysis, decision to publish, or preparation of the manuscript.

==============================
Extracellular polymeric substances (EPS) are essential for maintaining the structural integrity and function of biofilms. In this study, the influence of nutrient composition on biofilm formation and EPS production by Burkholderia pseudomallei was assessed using a quantitative assay after cultivation in brain heart infusion (BHI), Luria-Bertani (LB), and modified Vogel and Bonner medium (MVBM) for 2, 4 and 6 days. Biofilm biomass, the percentage of EPS in the biofilm matrix, and the biochemical composition of EPS were analyzed. The functional groups of EPS were characterized using Fourier transform infrared (FTIR) spectroscopy, while the microstructural features of the EPS were examined using scanning electron microscopy. We found that B. pseudomallei cultured in MVBM exhibited the highest biofilm biomass, despite having the lowest proportion of EPS within the biofilm matrix. In contrast, cultures in LB medium produced the lowest biofilm biomass but contained the highest percentage of EPS. These observations indicate that growth under high carbon/nitrogen (C/N) ratio, as in MVBM, favors biofilm biomass accumulation, whereas low C/N ratio conditions, such as LB and BHI, are associated with a high relative EPS content. Despite these differences, the carbon content of the EPS remained consistent across all media. Notably, the EPS derived from cultures grown in BHI, a nitrogen-rich medium, contained the highest protein content, which corresponded with noticeable amide peaks in FTIR spectra. Collectively, these findings enhance our understanding of how environmental C/N ratios influence B. pseudomallei biofilm development and EPS composition, with implications for bacterial persistence and adaptability in the environment.

Introduction

Burkholderia pseudomallei, the causative agent of melioidosis, is a soil-dwelling bacterium. Environmental factors such as soil texture, soil physico-chemical parameters including pH, salinity, moisture, nutrient content, and iron levels impact survival of the pathogen (Kamjumphol et al., 2015; Palasatien et al., 2008; Pongmala et al., 2022; Wang-Ngarm, Chareonsudjai & Chareonsudjai, 2014). The ability of this bacterium to form biofilm facilitates its persistence not only against antimicrobial agents (Mongkolrob, Taweechaisupapong & Tungpradabkul, 2015; Pibalpakdee et al., 2012; Sawasdidoln et al., 2010) but also against grazing by amoebae, as demonstrated in Acanthamoeba sp., a phagocytic organism in the environment (Bunma et al., 2023). The role of biofilms in protecting B. pseudomallei against antimicrobial agents, host immune responses and harsh environmental conditions has garnered significant attention, highlighting the need to better understand mechanisms of biofilm control. Recent NMR-based metabolomic analyses of B. pseudomallei biofilm and its extracellular polymeric substances (EPS), cultivated in LB and MVBM media, as demonstrated in our earlier work, have identified potential metabolic pathways that may serve as targets for controlling biofilm-associated infections (Srithabut et al., 2025).

The biofilm extracellular matrix, composed of water and EPS, primarily polysaccharides, proteins and DNA, that are fundamental of the establishment and maintenance of biofilm structure and properties of biofilm (Di Martino, 2018). These biomolecules support biofilm formation, protect cells against environmental stressors, facilitate adhesion to surfaces, and contribute to hydration and nutrient retention (Costa, Raaijmakers & Kuramae, 2018; Costerton et al., 1987; Flemming & Wingender, 2010; Wingender, Neu & Flemming, 1999). Microbial biofilm complexity arises from the dynamic interplay of physical and biological forces between cells and EPS within the extracellular matrix that shape biofilm characteristics (Wong et al., 2023). Moreover, many environmental factors such as pH, temperature, and oxygen concentration influence EPS production (Bayer et al., 1990; Jyoti, Soni & Chandra, 2024). The common type of EPSs produced by Burkholderia species, cepacian has been demonstrated to facilitate bacterial survival under various external stresses including stress, pathogenic interaction, desiccation and nutrient starvation (Ferreira et al., 2011).

The composition of EPS depends on the type of microorganism, with environmental factors largely determining variations in EPS production. A study in soil bacteria and fungi grown in a define medium supplemented with starch exhibited a higher EPS carbohydrate/protein ratio compared to those grown with glycerol supplementation. The elevated ratio was associated with the substrate complexity and correspond to increase hydrophobicity, which in turn conferred enhanced tolerance against water, solvents, and biocides (Oliva et al., 2025). The presence of sucrose in the culture conditions of dental plaque bacteria was demonstrated to influence both bacterial density and biofilm depth, as observed using confocal microscopy (Singleton et al., 1997). Furthermore, sucrose concentrations also impacted on biofilm composition including dry weight, bacterial counts and EPS content of Streptococcus mutans (Cai et al., 2016). The influence of the ratio of carbon to nitrogen (C/N) in the extreme halophilic archaeon Haloferax mediterranei using kinetic model analysis indicated that increasing nitrogen availability significantly increases EPS volumetric productivity and biomass (Cui, Shi & Gong, 2017). Several studies have demonstrated that the composition of growth media significantly influences the biochemical makeup of EPS, particularly the carbohydrate and protein content, in bacteria such as EPS excreted by lactic acid bacteria under static and dynamic conditions (Petrovici et al., 2017), carbon source optimization provided for fermentation increased the production of EPS and impacted EPS composition in Lactobacillus paracasei (Zhang et al., 2021).

Recent advances in Fourier transform infrared spectroscopy (FTIR) and scanning electron microscopy (SEM) have enabled precise characterization of EPS composition and structural architecture. FTIR identifies key functional groups such as O–H, C=O, and N–H, which are indicative of carbohydrates, proteins, nucleic acids, and lipids (Di Martino, 2018). The application of FTIR spectroscopy for characterization of pathogenic bacterial biofilms and EPS has been comprehensive reviewed (Chirman & Pleshko, 2021). Employing FTIR to investigate the impact of incubation temperatures and pH levels on biofilm structure of Salmonella enterica confirmed the alteration of organic compounds in the biofilm (Ariafar et al., 2019). More recently, combined FTIR, SEM and µ-Raman analyses have been applied to characterize monomicrobial biofilms of Pseudomonas aeruginosa and Escherichia coli (Paladini et al., 2024). Notably, when integrated with SEM, FTIR provides comprehensive insights into the structural properties of EPS, revealing how bacteria tailor their extracellular matrices in response to environmental conditions.

There are still some important gaps in our understanding of the influence of nutrient composition, in particular the availability of carbon and nitrogen, on biofilm development and the biochemical composition of EPS within B. pseudomallei biofilm in various environmental settings. Therefore, this study investigated how different culture media affect biofilm biomass, the proportion of EPS within the biofilm matrix, and the biochemical composition of EPS of B. pseudomallei ST-39, a strain previously isolated from soil in Khon Kaen, Thailand (Wang-Ngarm, Chareonsudjai & Chareonsudjai, 2014) and has been demonstrated to produce biofilm (Srithabut et al., 2025). Our findings provide new insights into the critical role of nutrient elemental composition, particularly carbon and nitrogen, in modulating biofilm development and EPS production B. pseudomallei.

Materials and methods

Culture media

The culture media for biofilm formation in this study included brain heart infusion (BHI) (Product No. M210-500G, Hi-media, India), an enriched medium with high carbon and nitrogen content (1.00% (w/v) proteose peptone, 1.25% (w/v) dehydrated calf brain infusion, 0.50% (w/v) dehydrated beef heart infusion, 0.20% (w/v) glucose, 0.50% (w/v) NaCl, 0.25% (w/v) Na2HPO4), Luria-Bertani (LB) (BDH), a common medium for bacterial cultivation (1.00% (w/v) tryptic soy broth, 0.50% (w/v) yeast extract and 1.00% (w/v) NaCl) and modified Vogel–Bonner medium, MVBM, a chemically defined medium with high carbon and low nitrogen availability (0.02% (w/v) MgSO4⋅7H2O (Merck), 0.20% (w/v) C6H8O7 (Kemaus), 0.35% (w/v) NaNH4HPO4⋅4H2O (Sigma, Singapore), 1.00% (w/v) K2HPO4 (Merck), 0.036% (w/v) CaCl2⋅2H2O (BDH), 2.00% (w/v) glucose (Kemaus)) (Kunyanee et al., 2016; Lam et al., 1980; Taweechaisupapong et al., 2005).

All chemicals used in this study were of analytical reagent grade or equivalent.

Bacterial strain and biofilm cultivation

An environmental B. pseudomallei ST-39 isolate (Wang-Ngarm, Chareonsudjai & Chareonsudjai, 2014) was used throughout this study. The bacterium from frozen stocks was streaked on Ashdown’s agar plates and incubated at 37 °C for 48 h. A single B. pseudomallei colony was grown in five mL of LB broth with agitation at 200 rpm (Innova 42R; New Brunswick Scientific, Edison, NJ, USA) at 37 °C overnight. The bacterial suspension was subsequently diluted to an OD550 ≈ 0.1 (UV-160 A, Shimadzu, Japan). One mL of diluted culture was inoculated into 99 mL of LB broth, resulting in a final concentration of 1% (v/v) inoculum. This culture was then further incubated to obtain an OD550 ≈ 1.0 (corresponding to ≈1 ×108 CFU/mL) as biofilm inoculum (Pakkulnan, Sirichoat & Chareonsudjai, 2024). One mL of the biofilm inoculum was added to 24 mL of fresh BHI, LB, or MVBM in 50 mL conical centrifuge tubes (Corning, Corning, NY, USA), prepared in quadruplicate for each media. Biofilm cultures were further incubated at 30 °C under static condition with vented lids for 2, 4, or 6 days, based on previously established growth and biofilm formation previously demonstrated (Kamjumphol et al., 2015; Wang-Ngarm, Chareonsudjai & Chareonsudjai, 2014). Biofilms cultures were harvested by centrifugation at 10,000× g for 15 min at 4 ∘C. Following removal of the supernatant, the pellets were washed three times with 25 mL phosphate-buffered saline (PBS). The pellets were then stored at −20 °C until further used. On the day of the experiment, the pellets were thawed at room temperature for 10 min and resuspended in 10 mL PBS. Subsequent analyses were performed to determine the total biofilm dry weight, extract EPS, and quantify total carbohydrate and protein concentrations.

Carbon to nitrogen (C/N) ratio in B. pseudomallei culture media

Supernatants from all three B. pseudomallei culture media on days 2, 4 and 6 were collected by centrifugation at 10,000× g for 15 min at 4 ∘C. Total organic carbon (TOC) left in the supernatant was determined followed the previous described (Walkley & Black, 1934) as originally described. The TOC was performed using the wet oxidation method. A 50 mL of the supernatant was transferred into a 250 mL Erlenmeyer flask and mixed with 10 mL of 0.1 M K2Cr2O7 solution. Subsequently, 20 mL of concentrated H2SO4 was then added, and the mixture was gently swirled for 1 min before being allowed to cool to room temperature. After cooling, 100 mL of distilled water, 10 mL o-phosphoric acid, and 0.2 g NaF were added, followed by the addition of 3–4 drops of diphenylamine indicator, which developed to violet-blue color (Nelson & Sommers, 1982). The mixture was then titrated with ferrous ammonium sulfate (FAS), prepared by dissolving 196.1 g Fe(NH4)2(SO4)2⋅6H2O and 20 mL concentrated H2SO4 in distilled water to a final volume of 1 L. The titration endpoint was identified by a color change from violet-blue to greenish blue.

The TOC concentration was calculated using the following equation: TOC%=Vb−Vs×N×0.003×100sample volume (mL)

where Vb is the volume of FeSO4 used for the blank (mL), Vs is the volume of FeSO4 used for the sample (mL), and N is the normality of K2Cr2O7 solution.

Total Kjeldahl nitrogen (TKN) was quantified according to the method previously described by Kirk (1950). A 200 mL of culture supernatant was digested with 50 mL of digestion reagent (0.77 M K2SO4, 0.029 M CuSO4, 2.46 M H2SO4) using a Kjeldahl Digester (Model 177550, Gerhardt) until the solution became clear. Following digestion, the mixture was cooled to room temperature, after which 200 mL deionized water, 0.5 mL phenolphthalein indicator, and 50 mL of mixed NaOH–Na2S2O3 solution (0.1 M each) were added. Nitrogen was subsequently distilled using a Gerhardt Distillation System (Model 7630, Gerhardt) under the following conditions: 0.5 min reaction time, 15 min distillation time, and 50% steam intensity. A total of 150 mL distillate was collected and titrated with 0.02 N H2SO4 to the endpoint, indicated by a color change from green to light purple. A blank, prepared using deionized water, was processed in parallel for correction.

The TKN concentration was calculated according to the following equation: TKN%=A−B×280×100sample volume (mL)

where A is the volume (mL) of H2SO4 used for titration of the sample, and B is the volume (mL) used for the blank.

C/N ratio was calculated using the following formula: C/N ratio=Total carbon content (g)Total Kjeldahl nitrogen content (g).

Measurement of B. pseudomallei biofilm biomass

Burkholderia pseudomallei biofilm biomass was determined as the total dry weight of B. pseudomallei using the volatile suspended solids (VSS) method (American Public Health et al., 2005). Briefly, glass fiber filter papers (GF/C, 1.1 µm pore size, four cm diameter; Whatman) were pre-dried at 105 °C for 1 h, cooled in a desiccator, and stored until use. For biomass collection, filters were placed on a Büchner funnel, pre-wetted with distilled water, and used to filter cell pellets that had been resuspended in 10 mL distilled water under vacuum. The retained biomass was subsequently rinsed with 10 mL of distilled water and subjected to vacuum filtration for additional 3 min. Filters containing the biomass were then dried at 105 °C for 1 h, cooled in a desiccator, folded, placed in pre-weighed crucibles, and weighed to obtain the initial dry mass (A). The crucibles were then ignited at 550 °C for 20 min until the content turned to gray ash, cooled in a desiccator, and reweighed to determine the post-ignition mass (B). The organic dry biomass was calculated as the difference between these two weights, with measurement performed using an analytical balance (Model MSE225S-100-DI; Sartorius, Germany).

Biofilm biomass was calculated using the following formula: Total dry weight of biofilm (mg/mL)=weight before ashing (mg) (A)−weight after ashing (mg) (B)×106Volume (mL).

EPS extraction

EPSs are classified into two main types based on their physical association with microbial cells: cell-bound EPS and soluble EPS. In this study, only cell-bound EPS was extracted from biofilm pellets using the cation-exchange resin (CER) method (Frølund et al., 1996; Jahn & Nielsen, 1995; Oliva et al., 2025; Oliva et al., 2024). Briefly, the frozen pellets were suspended in 10 mL PBS containing 1.6 g of CER Dowex (Dowex Marathon C sodium, strongly acidic, 20–50 mesh; Sigma, Singapore), then mixed thoroughly using a vortex mixer. The mixture was then shaken at 400 rpm for 1 h, after which the supernatant was collected. This was followed by centrifugation at 10,000× g for 15 min at 4 ∘C. The supernatant was transferred again to a fresh tube and centrifuged at 20,000× g for 20 min at 4 ∘C. EPS purification by monomer removal was performed via dialysis (MWCO 3,500 Da; T1 5015-46; Cellu.Sep) in deionized water (EPS:DI, 1:10) at 4 ∘C for 24 h. The retained EPS in the dialysis membrane was stored at −20 ∘C for subsequent analyses.

To determine the dry weight, functional groups, and morphological characteristics of the EPS, the EPS suspension was first concentrated using a rotary evaporator (Maxivac Beta, LaboGene) and stored at −70 ∘C for 24 h. It was then freeze-dried (MiniVac, LaboGene) to obtain a powdered sample, which was subsequently stored at −20 ∘C.

Measurement of B. pseudomallei EPS biomass

The biomass of extracted EPS was determined from 100 mL of culture and extrapolated to a per-liter yield. The EPS yield was expressed as the percentage of extracted B. pseudomallei EPS in biofilm grown in each culture medium. Percentage of EPS=Dry weight of EPS in biofilm (g/L)/Dry weight of biofilm (g/L)×100.

Biochemical composition and characterization of EPS

Total carbohydrate concentration in EPS

The total carbohydrate content in EPS was determined using the phenol-sulfuric acid method (DuBois et al., 2002), using glucose as the standard for generating a standard calibration curve. In brief, 0.5 mL EPS was mixed with 2.5 mL of 95–97% sulfuric acid (Merck) and 0.5 mL of 5% phenol (QReC), incubated for 10 min at room temperature, then heated in a 30 ∘C water bath for 15 min. After cooling for 5 min, the absorbance was measured at 490 nm using a double-beam spectrophotometer (UH5300; Hitachi). Carbohydrate concentrations were interpolated from the linear regression equation of the standard curve, and concentrations were adjusted for dilution factors.

Total protein concentration in EPS

The total protein concentration in EPS was determined using the Lowry method (Peterson, 1977) using bovine serum albumin as the standard for generating a standard calibration curve. An aliquot (0.5 mL) of EPS was mixed with 0.5 mL of Lowry reagent (4% Na2CO3, 0.2 M NaOH, 1% CuSO4, 2% NaNH4HPO4⋅ 4H2O (Sigma)) and incubated for 20 min. Then, 0.25 mL of Folin-Ciocalteu’s phenol reagent (Sigma F9252) was added, mixed thoroughly, and incubated for an additional 30 min. Absorbance was measured at 750 nm using a double-beam spectrophotometer. Protein concentrations were interpolated from the linear regression equation of the standard curve, and concentrations were adjusted for dilution factors.

Determining functional groups of EPS using FTIR

The functional groups of EPS were determined using FTIR (TENSOR 27, Bruker, Germany), following the method previously described by Bramhachari (Bramhachari et al., 2007) using potassium bromide (KBr) as a baseline control. EPS powder was finely ground with KBr to a final concentration of 0.01% (w/w) in a mortar. The mixed sample was placed in a mold and was compressed with a hydraulic press at 15,000 psi for 1–2 min. The sample was analyzed with an FTIR, and the spectra were interpreted to identify functional groups and chemical bonds, particularly OH-stretch compounds and N-compounds, using public databases, such as PubChem (https://pubchem.ncbi.nlm.nih.gov/), WebSpectra (https://webspectra.chem.ucla.edu/irtable.html), and ChemAnalytical (http://www.chemanalytical.com/services/ft-ir-spectra/).

Characterization of EPS using SEM

Structural characterization of EPS followed a previously described method (Schrand et al., 2008). Briefly, the extracted EPS samples were desiccated at −20 ∘C for 24 h. Subsequently, the dried samples were mounted on stubs using carbon adhesive tape and then sputter-coated with a gold layer approximately 10–20 nm thick prior to imaging with a scanning electron microscope (SEC, SNE-4500M).

Statistical analysis

Data was obtained from three independent experiments, each with three replicates (n = 9). Statistical analysis was performed using GraphPad Prism version 10 (GraphPad Software, Boston, MA, USA). Comparisons of the total dry weight of biofilms and EPS, carbon-to-nitrogen ratios, carbohydrate and protein concentrations, were performed using one-way ANOVAs followed by Tukey post-hoc test for comparison between pairs. The levels required for statistical significance were *p < 0.05, ∗∗p < 0.01 and ∗∗∗ p < 0.001. Bar graphs in the figures display the mean values with error bars representing the standard deviation (SD). Linear regression analysis was performed to analyze correlation of C/N ratio of media on the biofilm and EPS dry weight.

Results

Effect of culture medium on biofilm biomass, EPS dry weight and the EPS yield in B. pseudomallei biofilm

The dry weight (the total dry mass) of B. pseudomallei biofilms grown in BHI, LB and MVBM increased throughout the course of observations (Table 1). The dry weight of biofilm was highest in cultures grown in MVBM, followed by those cultivated in BHI and LB. On day 6, the biofilm dry weight in MVBM was significantly greater than that observed in both BHI and LB cultures (p < 0.001). In contrast, biofilms formed in LB exhibited the lowest biofilm biomass among the three media.

Table 1 Dry weight biofilms of B. pseudomallei determined by the VSS method and EPS biomass of B. pseudomallei grown in BHI, LB and MVBM media at 30 °C on days 2, 4 and 6.

Culture media	Dry weight of biofilm (g/L)	Biomass of EPS (g/L)	
	Day 2	Day 4	Day 6	Day 2	Day 4	Day 6	
BHI	0.3365 ± 0.0142	0.4099 ± 0.0022	0.4397 ± 0.0420	0.1571 ± 0.0072	0.2347 ± 0.0011b	0.2638 ± 0.0029c	
LB	0.2385 ± 0.0084a	0.2644 ± 0.0021a	0.3475 ± 0.0036a	0.1595 ± 0.0001	0.1879 ± 0.0004	0.2500 ± 0.0030	
MVBM	0.3832 ± 0.0186	0.4155 ± 0.0046	0.5301 ± 0.0057b	0.1370 ± 0.0114	0.1788 ± 0.0005	0.2561 ± 0.0005c	
Notes.

Data are shown as mean ± SD from three independent experiments, each performed in triplicate (n = 9).

a p < 0.001 significant lower on the same day withing different culture media.

b p < 0.001 significant higher on the same day within different culture media.

c p < 0.01 significant higher on day 6 compared to day 2 within same culture media.

The dry weight of EPS extracted from B. pseudomallei biofilms increased over the experimental period in all media (Table 1). On day 4, the EPS dry weight monitored in BHI medium was significantly higher than those of the other two media (p < 0.001). The dry weight of EPS in both LB and MVBM media increased significantly from day 2 to day 6 (p < 0.01).

The percentage of EPS relative to total biofilm biomass was analyzed (Fig. 1). Notably, the LB medium consistently produced the highest EPS percentage across all time points throughout the incubation period, in comparison to both BHI and MVBM media (p < 0.001). The MVBM medium demonstrated the lowest EPS percentage within the biofilm matrix (p < 0.001).

Figure 1 Percentage of EPS relative to the biofilm biomass of B. pseudomallei grown in BHI, LB and MVBM media, incubated statically at 30 °C for 2, 4 and 6 days.

Data are represented as mean ± SD from at least three independent experiments, each performed in triplicate (n = 9). Asterisks denote statistical significance (∗∗∗p < 0.001).

The carbon and nitrogen (C/N ratio) of the nutrients directly influence bacterial metabolism, growth strategy, and biofilm development. Therefore, the C and N contents were measured in freshly prepared BHI, LB and MVBM media as well as in spent media collected after biofilm cultivation on days 2, 4 and 6 (Table S1). The corresponding C/N ratios were calculated (Fig. 2). The C/N ratios of fresh BHI, LB and MVBM were 8.5, 12 and 98, respectively. The C/N ratios in MVBM following 2, 4, and 6 days of incubation were 190, 359 and 417, respectively, the highest in any of the culture media (p < 0.001) according to the low N content. BHI and LB exhibited relatively low C/N ratios of 11–22 and 11–16, respectively. These findings suggest that MVBM contains excess carbon and limited nitrogen, resulting in the observed extremely high C/N ratio. No significant difference was observed between the C/N ratios of BHI and LB (p > 0.05).

Figure 2 Carbon to nitrogen ratios in BHI, LB and MVBM culture media on days 0 (fresh medium), 2, 4 and 6 following cultivations at 30 ° C.

Supernatants were collected by centrifugation at 10,000×g for 15 min at 4 ° C and analyzed using the Walkley & Black (1934) method. Data are represented as mean ± SD from at least three independent experiments, each performed in triplicate (n = 9) . Asterisks denote statistical significance (∗∗ p < 0.01 and ∗∗∗p < 0.001).

Linear regression analysis was used to investigate the correlation of the C/N ratio of BHI, LB and MVBM media with the dry weight and percentage of EPS in the biofilm matrix of B. pseudomallei measurement on days 2, 4 and 6. As the C/N ratio changes over time, we found a moderate positive correlation between biofilm dry weight and C/N ratio of MVBM medium (R2 = 0.5570) (Fig. 3A) and a strong positive correlation between percentage of EPS in biofilm matrix and the C/N ratio of MVBM medium (R2 = 0.7131) (Fig. 3B), indicating the correlation of C/N ratio of MVBM medium to dry weight biofilm and percentage of EPS in B. pseudomallei biofilm matrix but not in the case of BHI and LB medium (data not shown).

Figure 3 A moderate positive correlation was observed between C/N ratio of MVBM medium to (A) biofilm dry weight and a strong positive correlation of (B) percentage of EPS in B. pseudomallei biofilm matrix.

Purple, red, and blue dots represent correlation observed at 2, 4, and 6 days of B. pseudomallei cultivation, respectively.

Effect of culture medium on biochemical composition of B. pseudomallei EPS

The biochemical composition of the EPS extracted from B. pseudomallei biofilms varies according to the cultivation medium. The total carbohydrate content in EPS showed a progressive increase in all media, with a substantial elevation observed by day 6 (Fig. 4A). In contrast, nitrogen content, representing protein quantity, was highest in biofilms grown in BHI medium, followed by those in LB and MVBM media (Fig. 4B). These results suggest a critical influence of nutrient composition on EPS profiles, with BHI medium preferentially supporting protein synthesis, while all tested media facilitated time-dependent carbohydrate accumulation.

Figure 4 Biochemical composition of B. pseudomallei. EPS produced in BHI, LB and MVBM media under static incubation at 30 ° C for 2, 4 and 6 days.

(A) Total carbohydrate in EPS analyzed using the phenol-sulfuric acid method and (B) total protein in EPS analyzed using Lowry method. Data is represented as mean ± SD from at least three independent experiments with three replicates each (n = 9). Asterisks denote statistical significance (∗ p < 0.05, ∗∗ p < 0.01 and ∗∗∗p < 0.001).

FTIR spectroscopy analysis of EPS

FTIR spectroscopy analysis was employed to characterize the functional groups present in the EPS extracted from 6-day-old B. pseudomallei biofilm grown in BHI (Fig. 5A), LB (Fig. 5B), and MVBM (Fig. 5C). The EPS consisted primarily of carbohydrates, proteins and lipids, with both similarities and subtle differences among the media. Notably, EPS produced in LB and MVBM media was rich in carbohydrate-associated functional groups, whereas EPS from BHI medium exhibited comparable levels of carbohydrate and protein functional groups. The characteristic peaks associated with functional groups observed including broad O–H and/or N–H stretching vibrations around 3,400–3,200 cm−1 and prominent amide C=O stretch (1,700–1,600 cm−1) bands corresponding to polysaccharide (O–H stretching) and protein components (N–H and C=O stretching), respectively. Notably, the EPS extracted from bacteria grown in BHI and LB medium exhibited the highest peak intensities, particularly in the amide (C=O stretch (1,700–1,600 cm−1)). These findings indicated that the different C and N components in each culture medium impacted on the biochemical composition of bacterial EPS by day 6. However, on days 2 and 4, the functional group compositions of EPS were similar in LB, MVBM, and BHI (data not shown). These observations were consistent with the results obtained from the quantitative analysis of total carbohydrate and protein concentrations in EPS. Additionally, lipid-associated functional groups were also detected across all media.

Figure 5 FTIR Spectra of EPS extracted from 6-day-old B. pseudomallei biofilms cultured in (A) BHI, (B) LB, and (C) MVBM media.

Y- and X-axis represent the percentage of transmittance and wave number (cm−1), respectively.

Scanning electron microscopic structure of the B. pseudomallei EPS

SEM images revealed the EPS architecture extracted from the biofilms of B. pseudomallei cultivated in BHI, LB and MVBM media for 2, 4 and 6 days (Fig. 6). On day 2, EPS from all three media was smooth in morphology but had a more granular appearance on days 4 and 6. EPS from BHI cultivation appeared as tightly packed clusters while EPS from LB and MVBM cultures demonstrated large and compact aggregates with irregular and coarse surface morphology.

Figure 6 SEM images of EPS extracted from the B. pseudomallei biofilms grown in LB, MVBM and BHI media on days 2, 4 and 6.

The EPS structure was observed at 7,000×magnification.

Discussion

Biofilm formation by B. pseudomallei is essential for its persistence in diverse environments and contributes to its pathogenic potential. EPS are essential constituents of bacterial biofilms, contributing significantly to their structure and function. Nutrient availability plays a pivotal role in modulating biofilm development, structural integrity, and the composition of EPS. Elucidating how specific nutrients influence these parameters is crucial for advancing our understanding of biofilm biology. This study examined the effects of three bacterial culture media, BHI, LB and MVBM on EPS production and biochemical composition of B. pseudomallei biofilms. The results demonstrated that variations in C/N ratio across the media differently impacted EPS yield and composition. Cultivation in MVBM, with an extremely high C/N ratio, resulted in the highest biofilm biomass but the lowest EPS proportion within the biofilm matrix. In contrast, the lower C/N ratios in LB and BHI promoted a higher EPS percentage. Notably, EPS derived from BHI cultivation, a nitrogen-rich medium, exhibited prominent amide peaks in FTIR spectra, indicating elevated protein content.

The total weight of B. pseudomallei biofilm reflects overall biofilm growth, including bacterial cells, water, EPS, and extracellular DNA (eDNA) (Pakkulnan et al., 2019). In this study, B. pseudomallei exhibited the highest biofilm biomass in MVBM, likely due to the glucose content and elevated C/N ratio of this medium. The presence of 2% glucose may also induce osmotic stress, potentially activating regulatory pathways associated with biofilm development, as observed in other bacterial species (Li et al., 2019; Szczesny et al., 2018). Consistent with these findings, previous studies have also reported that MVBM promotes more robust B. pseudomallei biofilm formation than BHI (Taweechaisupapong et al., 2005) and LB (Anutrakunchai et al., 2015).

EPS is crucial for biofilm structure and function, providing protection against mechanical and chemical stresses, facilitating nutrient retention and microbial community interactions (Atmakuri et al., 2024; Flemming & Wingender, 2010). This study demonstrated that lower C/N ratios of LB and BHI favor the higher proportion of the cell-bound EPS in B. pseudomallei biofilm compared to the very high C/N ratio of MVBM. This finding highlights the importance of maintaining an optimal C/N balance to achieve maximum EPS production (Atmakuri et al., 2024). We found the highest proportion of EPS in biofilm grown in LB, which is consistent with a study of EPS from Bacillus amyloliquefaciens when supplemented with 1% yeast extract as a nitrogen source, compared to ammonium chloride, tryptone, phenylalanine (Rao et al., 2013). Nonetheless, it is important to acknowledge that the carbohydrate content inherent in yeast extract may serve as a confounding factor, potentially contributing to the enhanced EPS yield observed in their study.

The high C/N ratio observed in fresh MVBM medium was attributed to its low nitrogen content, which increased during the 6-day observation period, stands in contrast with the unchanging C/N ratios in LB and BHI. This may be influenced by free EPS which can be released during biofilm production in media containing high carbohydrates, as observed in Lactobacillus plantarum (Tallon, Bressollier & Urdaci, 2003), Escherichia coli (Eboigbodin & Biggs, 2008; Han, Enfors & Häggström, 2003) and B. subtilis (Omoike & Chorover, 2004). However, the depletion of nitrogen from the medium should also be considered as a factor influencing changes in the C/N ratio. Furthermore, in E. coli cultures grown in LB, the protein content of free EPS increased, whereas carbohydrate content rose only when the LB medium was supplemented with glucose (Eboigbodin & Biggs, 2008). EPS production in K. pneumoniae, P. aeruginosa, and B. cepacian was optimized by using ammonium sulfate as the nitrogen source and glucose as the carbon source, with maximum yields obtained after eight days of incubation (Jyoti, Soni & Chandra, 2024). This is consistent with our study, which maintains a constant C/N ratio during B. pseudomallei cultivation in LB medium. Additionally, the observed correlation between the high C/N ratio of the MVBM medium and lower percentage of EPS observed in this study aligns with the finding of Cui, Shi & Gong (2017), who demonstrated that EPS productivity by H. mediterranei in a controlled fermentation model exhibited an inverse correlation with the C/N ratio when defined over as set cultivation period. Nevertheless, changes in the C/N ratio could also result from uncontrolled cell lysis and/or the release of intracellular storage polymers, such as polyhydroxybutyrate (PHB), into the medium.

The natural ecosystem comprises microbial community composition and a diverse of organic and inorganic materials, including nitrogen and carbon sources, which are essential for microbial metabolism and productivity (Gebert et al., 2025). A vertical soil study found that deeper soil, which has low nitrogen (high C/N), makes it easier for extracellular enzymes to help accumulate EPS (Feng et al., 2026). Therefore, regarding this study, in carbon-rich soils with high C/N ratios, B. pseudomallei may prioritize EPS formation, whereas in nitrogen-rich environments, the bacterium may favor increased biomass production. These findings highlight the critical influence of nutrient availability on the adaptations and persistence of B. pseudomallei, mediated through EPS modulation during its survival both in the host and the soil environment. A deeper understanding of EPS and biofilm formation by B. pseudomallei may be gained in the future to aid in predicting and limiting the persistence of this pathogen in the environment.

Biochemical analyses have consistently established that carbohydrates and proteins constitute the predominant components of bacterial EPS (Tsuneda et al., 2003). These macromolecules are crucial in mediating EPS functionality, structural stabilization of the biofilm matrix, maintaining of hydration, nutrient storage, and protection against environmental stresses (Flemming & Wingender, 2010). In this study, the highest total carbohydrate concentration was observed on day 6, with all cultivation media exhibiting comparable carbohydrate levels at that time point. This finding indicates that carbohydrates are the major constituent of EPS. Our study provides further evidence of the carbohydrate composition of the exopolysaccharide in B. pseudomallei biofilm (Mangalea, Borlee & Borlee, 2017). Notably, the carbohydrate composition of the exopolysaccharide in B. pseudomallei biofilms has been shown to play a critical role in forming a diffusion barrier that impedes antibiotic penetration (Mongkolrob, Taweechaisupapong & Tungpradabkul, 2015).

In comparison, BHI supported the highest protein content of B. pseudomallei EPS, followed by LB, likely to be associated with the rich nitrogen source in the medium. This is in good agreement with the highest yield of EPS from B. amyloliquefaciens when yeast extract was supplied as the nitrogen source (Rao et al., 2013). However, despite BHI facilitating the highest total protein content in B. pseudomallei EPS, it exhibited a lower proportion of EPS relative to biofilm biomass when compared to LB medium. The lower proportion of EPS in biofilm is contrasts with findings in Bacillus cereus, where BHI, a nutrient-rich medium, supported the highest level of exopolysaccharide production compared to cultures grown in LB or trypticase soy broth (Minimol et al., 2019).

Distinct chemical composition in the EPS produced by B. pseudomallei in three different culture media were confirmed by FTIR spectroscopy. All samples contained common functional groups, including O–H stretching vibrations, indicative of carbohydrates (Di Martino, 2018). A previous study reported that B. pseudomallei EPS contains carbohydrates such as maltose and glucose, as detected using NMR-based metabolomics when cultured in LB and MVBM, respectively (Srithabut et al., 2025). However, the spectral intensity varied depending on the medium. For instance, the EPS from BHI and LB media exhibited strong amide C=O stretching bands, which are characteristic markers of proteins. These functional groups are also commonly found in EPS produced by P. fluorescens when cultured in LB (Delille, Quiles & Humbert, 2007), B. vallismortis (Ding et al., 2018), E. coli culture in LB (Eboigbodin & Biggs, 2008) and Enterobacter sp. (Li et al., 2021). The presence of these functional groups in LB-derived EPS suggests that both BHI and LB support the formation of protein-rich EPS, likely due to their nitrogen-rich components. However, Minimol et al. (2019) employed FTIR to characterize exopolysaccharide from B. cereus, showing that exopolysaccharide produced in BHI medium displayed typical polysaccharide features, including peaks indicative of carbohydrates and glycosidic linkages. Our study provides additional support for the findings of FTIR wave number represent polysaccharides, amide (C=O stretch) with the in situ monitoring of nascent P. fluorescens biofilm cultured in LB medium using attenuated total reflectance-FTIR (Delille, Quiles & Humbert, 2007).

The SEM images of B. pseudomallei EPS exhibited slightly distinct morphological characteristics across the three culture media. Initially with a smooth appearance, the EPS progressively developed granular structures over time, particularly in BHI. Our recent study suggested that these morphological alterations may result from differences in metabolite composition arising from varying nutrient availability in LB and MVBM (Srithabut et al., 2025). Subtle differences in EPS features may be caused by the carbon and nitrogen content in tested media, which in turn may affect its functional roles, including its capacity to serve as a cohesive matrix component and provide protection against antimicrobial agents and the host immune system. However, the preparation steps used for EPS extraction and SEM sample processing may cause collapse and distortion of hydrated EPS. Further studies employ complementary imaging approaches using cryogenic SEM method (Dohnalkova et al., 2011) or confocal laser scanning microscopy (Lu et al., 2024) to better preserve and characterize the native architecture of EPS in B. pseudomallei biofilms.

To the best of our knowledge, this is the first comprehensive study to systematically compare the influence of BHI, LB and MVBM media on biofilm formation, EPS production, and biochemical characteristics of B. pseudomallei, providing new insights into how medium composition shapes these properties. Based on our findings, it is theoretically plausible to guide future biofilm research using tailored media composition. MVBM medium, characterized by a high C/N ratio, is suitable for promoting high biofilm biomass with a low EPS proportion, whereas LB and BHI media are more appropriate for enhancing EPS production. Our study shows that C/N ratios affect not only the quantity of biofilm and EPS in B. pseudomallei, but also EPS composition. While both contribute to survival under antimicrobial, immune, and environmental stress, they serve distinct roles: EPS provides adhesion, protection, and a physical barrier, whereas biofilms, as structured communities, promote long-term persistence. Thus, although biofilms may play the more dominant role in persistence, EPS remains essential for their formation and function.

We are aware of the study’s limitations. Confounding factors, such as pH, temperature, nutrient availability, and oxygen can affect EPS production (Bayer et al., 1990; Jyoti, Soni & Chandra, 2024), however, this was conducted under a single condition. Moreover, the EPS analysis was restricted to cell-bound fractions of B. pseudomallei biofilms and did not account for the soluble EPS. We acknowledge that the risk of intracellular contamination during EPS extraction may occur, and we suggest that future studies include the measurement of intracellular enzyme markers (e.g., lactate dehydrogenase) and DNA leakage to verify that the extracted EPS fraction is not confounded by cell lysis. Although triple washing of the biofilm cultures, traces of medium-derived macromolecules may remain, potentially influencing biochemical analyses of EPS. In addition, comparative FTIR spectrum for each media in the absence of bacteria were not performed. Consequently, the findings may not fully reflect the impact of culture media components to overall EPS properties of B. pseudomallei biofilm.

Conclusions

This study provides important insights into how different culture media influence the growth, EPS production, and biochemical composition of B. pseudomallei ST-39 biofilm. MVBM, an inorganic minimal medium with a high C/N ratio, supported the highest biofilm biomass, likely due to its nitrogen limitation and carbon abundance. In contrast, nutrient-rich BHI and LB promoted higher levels of protein synthesis within EPS. FTIR analysis confirmed that each medium had a biochemically distinct EPS: BHI and LB promoted protein-rich EPS, whereas MVBM favored carbohydrate-rich structures. Furthermore, SEM imaging showed that EPS structures were more compact and well-developed in BHI medium. However, the extensive processing steps involved in sample preparation may have altered these structures, meaning they might not accurately reflect their native form. These findings highlight the critical influence of nutrient availability on the adaptations and persistence of B. pseudomallei, mediated through EPS modulation during its survival both in the host and the soil environment.

Supplemental Information

Supplemental Information 1 The data were calculated from triplicates within a single experiment, and the experiments were performed on three independent occasions (total samples, n = 9)

Supplemental Information 2 FTIR absorption bands and corresponding functional groups identified of EPS extracted from B. pseudomallei biofilms grown in BHI, LB and MVBM media for 6 days

We would like to acknowledge Prof. David Blair for editing the manuscript via the university’s Publication Clinic. University, Thailand.

Additional Information and Declarations

Competing Interests

Author Contributions

Data Availability

The authors declare there are no competing interests.

Suthantip Srithabut conceived and designed the experiments, performed the experiments, analyzed the data, prepared figures and/or tables, authored or reviewed drafts of the article, and approved the final draft.

Thotsapol Chaianunporn conceived and designed the experiments, authored or reviewed drafts of the article, and approved the final draft.

Sorujsiri Chareonsudjai conceived and designed the experiments, performed the experiments, analyzed the data, prepared figures and/or tables, authored or reviewed drafts of the article, and approved the final draft.

Pisit Chareonsudjai conceived and designed the experiments, performed the experiments, analyzed the data, prepared figures and/or tables, authored or reviewed drafts of the article, and approved the final draft.

The following information was supplied regarding data availability:

The raw measurements are available in the Supplementary Files.

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
