# Peer review of "Impact of bacterial culture medium on composition and characteristics of Burkholderia pseudomallei extracellular polymeric substances"

_PeerJ, doi:10.7717/peerj.20488_

## Round 0.1 · original submission · Major Revisions

· Academic Editor

Major Revisions

After careful consideration of the comments raised by reviewers, I invite you to revise your work to improve content and clarity as suggested by both reviewers.

Reviewer 1 ·

Basic reporting

The authors compared and analysed the characteristics of EPS of B. pseudomallei cultivated in different medium. Overall, the authors did a massive and detailed analysis in this study. However, there are some additional information required to better understand the manuscript.

Experimental design

The authors use one environmental strain in this study and compared the biofilm formation at three different time points for three different media.
Methods
1)The authors may add appropriate references in the methods section.
2)Is the strain used an originally high biofilm producing strain? Please specify.
3)How are the time points selected?
4)Are the pH, temperature and other factors optimised in this study?
5) Please include the method for biofilm formation, cite appropriately.
6)The authors may provide formula for the all the calculations.
7) Why only bound EPS was extracted and used in the study? How about soluble EPS? Both involve in the formation of biofilm.

Validity of the findings

The authors may need to rephrase and restructure the paragraph in results. Some details are missing.
1) Line 203: What does "increased in similar patterns" referring to?
2) Line 209: The EPS content referring to percentage of EPS or dry weight? Please use the same term for all to avoid confusion.
3)It is recommended to include only percentage of EPS in figure and the other two parameters: dry weights in table format.
4)Line 215: What does "nutrient balance" referring to. Please define.
5)Please add brief description in the figures' legend.

Discussion
The authors are trying to relate the C/N ratio with environmental conditions but there is limited discussion on this. Please improve on the discussion structure and provide appropriate citations.

Conclusion
The conclusion is lengthy.

·

Basic reporting

With regard to 'Basic Reporting', some issues have been identified with clarity, literature references, figures, and raw data availability.

Therefore, I recommend that the following points be addressed.

1. Introduction

Line 44 - 45: The references included do not explicitly address how "soil texture" impacts pathogen survival. Please provide a supplementary reference, (consider statements in https://www.nature.com/articles/s41598-022-12795-0)

Line 48: The authors state that biofilm formation facilitates survival "within Acanthamoeba sp.". The study referenced (Bunma et al., 2023) compares wild-type strain H777 to mutant strain M10. However, in Figure 2, Bunma et al., demonstrate that H777 and M10 have similar survival within Acanthamoeba sp. Please rephrase this to accurately reflect the conclusions of Bunma et al., regarding amoeba grazing.

Line 51 - 54: The authors here refer to "Recent NMR-based metabolomic analyses". It should be made clear here that the study the authors refer to was in fact performed by their own group.

Line 56: The authors state that polysaccharides and proteins "constitute 75 - 89% of EPS". Only a singular reference provided, Tsuneda et al., (2003), supports this statement. Furthermore, the study conducted by Tsuneda et al., is highly specific to the 27 heterotrophic bacteria that were isolated from a specific structure within a wastewater treatment plant. It is not accurate to imply that the "75 - 89%" statistic would be applicable to all bacterial EPS, or the EPS of Burkholderia. The majority of biofilm studies do not provide an exact percentage of components within a biofilm, as these are likely context, species, and experiment specific. Please change this.

Line 57: The reference to "Patrick Di 2018" is incorrect.
In the reference section, Lines 453 - 455 list the citation:
"Di Martino P. 2018. Extracellular polymeric substances, a key element in understanding biofilm phenotype. AIMS Microbology 4:274-288. https://doi.org/10.3934/microbiol.2018.2.274";.
However, in Lines 535 - 536, the authors list the citation:
"Patrick Di M. 2018. Extracellular polymeric substances, a key element in understanding biofilm phenotype. AIMS Microbiology 4:274-288. 10.3934/microbiol.2018.2.274".
Please remove this duplicate reference, and ensure the correct format is used.

Line 58 - 60: The references provided as supporting evidence do not explicitly describe the biofilm as supporting "nutrient retention". Please source an appropriate reference for this claim.

Line 65: The authors state that the diverse EPSs produced by Burkholderia species provide a survival advantage under various conditions. The provided reference to Ferreira et al., (2011) is primarily concerned with only "cepacian", and does not describe other diverse EPSs. Please provide additional evidence to support this claim.

Line 70 - 71: The statement here that "cultures grown in starch media exhibit a higher EPS carbohydrate-to-protein ratio compared to those grown in glycerol" requires further explanation. The study cited (Oliva et al., 2025), used a defined media that contained yeast extract and peptone, in addition to the carbon sources stated.

Line 72: The authors state that "Similarly, under carbon-rich conditions, sucrose supplementation leads to significantly denser and thicker oral biofilms".
I cannot find any support for this statement within Kolenbrander & London (1993).
The second reference, Sutherland (2001), is a review. This review also (incorrectly) cites Kolenbrander & London (1993).
The correct source of this information is actually:
Singleton S, Treloar R, Warren P, Watson GK, Hodgson R, Allison C. Methods for Microscopic Characterization of Oral Biofilms: Analysis of Colonization, Microstructure, and Molecular Transport Phenomena. Advances in Dental Research. 1997;11(1):133-149. doi:10.1177/08959374970110010401
Please amend the reference to reflect this. For a more recent exploration of this, see https://journals.plos.org/plosone/article?id=10.1371/journal.pone.0157184.

Line 74: This sentence should be rephrased for clarity. The EPS referred to by Cui et al., (2017) is produced by Haloferax cultures in well-mixed, aerated batch cultures. The EPS referred to by the authors has previously been limited to cultured biofilms.

Line 79: The authors state that EPS is affected by growth media composition. The reference to Liu et al., (2010) describes an "intracellular bioflocculant" produced by Chryseobacterium daeguense primarily after cell death. This study is not relevant to EPS produced during biofilm formation.

Line 80: The study by Jyoti et al., (2024) refers to "Burkholderia cepacian". This is an error. The actual name of this strain is "Burkholderia cepacia, strain UTJ501". Furthermore, the strains of Klebsiella, Pseudomonas, and Burkholderia were isolated from wastewater by Jyoti and grown in consortia for biofilm assessment. Because these strains are mixed, it cannot be definitively stated which bacterial strain is being influenced by the change in growth media. I strongly recommend using alternative references for these claims.

Line 88: Please provide an example of where FTIR has been used for assessing biofilm structural properties.

Line 96: Please rephrase "nutrient quality". Nutrient density or concentration may be more accurate.


2. Materials and Methods
• Please put brackets around all abbreviations.
• Relative centrifugal force units (g) needs to be in italics

3. Results (Figures)

• There is no inclusion of A, B, and C in the JPEG provided for Figure 1.
• FIG 1C: Please edit the y-axis title to more accurately reflect your analysis. Currently it states, “Percentage of EPS in biofilm”. However, according to the figure legend and text, this should read “Percentage of EPS relative to biofilm biomass”.
• This data is better presented as a scatter plot so that the individual data points are visible and can be assessed.

• Lines 248 – 249: Please rephrase. The total “carbon” content in EPS was not measured by the phenol-sulfuric acid method. This method only measures total carbohydrate content.


4. Discussion

Line 305 - 306: The Anutrakunchai et al., 2015 citation is sufficient here.

Line 313 - 315: The authors reference a study by Rao et al., (2013). In this study, Rao et al., assessed the effect of different nitrogen sources on the EPS production of Bacillus amyloliquefaciens. Nitrogen sources were tested at "1%". The sources were ammonium chloride, tryptone, phenylalanine, and yeast extract. Yeast extract is also a source of complex carbohydrates, confounding this experiment.

Line 343 - 346:
Please state the full name of "B. cereus" the first time it is used.
In the study by Minimol et al., (2019), "EPS" is defined as "exopolysaccharide".
However, in the present study, the author uses EPS to refer to all extracellular polymeric substances.
Please clarify this paragraph, by stating that in Bacillus cereus, Minimol et al., found that BHI media supported higher exopolysaccharide production.

Line 350 - 352: The composition of Burkholderia extracellular polymeric substances has long been known to contain carbohydrates - prior to the NMR method used by the authors.
See Mongkolrob et al., (2015), https://onlinelibrary.wiley.com/doi/full/10.1111/1348-0421.12331. Mangalea et al., (2017), https://link.springer.com/article/10.1007/s40475-017-0118-2#citeas

Line 360 - 363: The author states that in contrast to their results, where "strong amide C=O stretching bands" were detected, EPS extracted by Minimol et al., from BHI medium displayed typical polysaccharide features in FTIR spectra.
However, as stated, Minimol et al., extracted exopolysaccharide only. Therefore, it makes sense that features indicative of glycosidic linkages were found. The use of EPS for two different terms has caused confusion.

Experimental design

With regard to 'Experimental design' there are issues with the level of information and detail provided in the methods.

To address these issues, I recommend the following:


1. MATERIALS & METHODS

Line 101: The composition here is too non-specific. If BHI broth was purchased from Hi-media, please state the specific product number. How were these concentrations determined? Please state percentage concentrations in (w/v).

Line 105 - 109: Please provide the original source of the recipe for Modified Vogel-Bonner Medium (MVBM). Other recipes for MVBM use D-gluconate as a carbon source.

Lines 119 - 120: This sentence is unclear. It may be misinterpreted.
Was the diluted culture (OD550 = 0.1) first made and then used to inoculate a new culture at 1% of the original concentration?
To improve clarity, please state the exact volume used to inoculate the new culture.
For example “100 uL of diluted culture was then used to inoculate 10 mL of LB broth, resulting in a final concentration of 1% (v/v). This culture was then incubated for a further x h to obtain an OD550 = 1.0”

Line 120: How were bacterial cells “washed”.
Was the entire 10 mL culture first pelleted via centrifugation?
Was the supernatant discarded?
What volume of PBS was the pellet suspended in? Was it 5 mL? If not, then you are concentrating the bacteria.

Line 121 - 122: This methodological step is unclear.
What volume was used to suspend the pellet?
Was the bacterial pellet suspended in PBS and then used to inoculate the LB, MVBM, or BHI at 1% (v/v)?
Or, was the bacterial pellet directly suspended in LB, MVBM, or BHI, then used to inoculate a secondary culture of LB, MVBM, or BHI at 1% (v/v)?
If the final volume of the biofilm culture is 25 mL, were they inoculated with 250 uL bacterial suspension?
Please state what size/brand conical tube was used. Were the conical tubes sealed or vented?

Line 131 - 133: Measurement of Total Organic Carbon (TOC) via method of Walkley & Black (1934). The Walkley and Black method for determining total organic carbon is primarily used for measuring soil organic carbon. Medical microbiologists may be unfamiliar with this process.
Were any adaptations required for performing this method on culture supernatant?
Can you provide a reference to this method being used for a similar purpose?

Line 133 - 134: Total Kjeldahl nitrogen. A brief explanation of the Kjeldahl method here would be useful for microbiologists. Again, many will be unfamiliar with this method, as it is primarily used in industry for determining nitrogen and protein concentration.

Line 139 - 140: Volatile suspended solids (VSS) method. The authors should be aware that many readers of this manuscript will be unable to access the method referenced in Standard methods for the examination of water & wastewater, 2005. Therefore, this method needs to be explained in more detail. Other VSS methods I have been able to access are highly detailed and technically complex.
Please include the following information:
Was the original pellet stored at -20 C used in its entirety? If not, how was it thawed and divided?
What wet weight of cell pellet was used?
Were samples evaporated to dryness initially?
How many times were samples heated and cooled?
Was the dried biomass ignited? How long for? What instrument was used for ashing?
Has this process been used to measure biofilm biomass from other pure cultures of bacteria?


Line 146 - 149: EPS Extraction – Cation-exchange resin. Please provide more information.
What volume of PBS was used to suspend the pellet?
Was there any attempt to control for cell lysis during this process?
Please include reference to a more recent publication that has used the Dowex method.

Line 164: Please rephrase. Was 1 L of culture used? 1 L of biomass?

Line 188: Please state whether percentage concentration was (w/v) or (v/v)

Line 196 - 197: Characterization of EPS using SEM. The citation included here Schrand, 2005 is an extended abstract of a paper presented at a conference. It does not adequately describe a method for preparation of EPS for SEM. Please provide an alternative reference for this method.


2. RESULTS

Effect of culture medium on biofilm biomass, EPS dry weight and the EPS yield in B. pseudomallei biofilm
• The measurement of biomass was calculated using the VSS method. It may be useful to refer to this method in the figure legend.
• Some of these statistical differences, while significant, are not useful. There is no experimental reason to compare the Dry weight of BHI biofilm at Day 2 with the Dry weight at Day 6. The hypothesis is that differences in media accounts for differences in biofilm formation, therefore the only relevant comparisons are between media.
• Having an excess of asterisks and lines between bars risks confusing the reader
• Line 221 – 222: Please include exact quantities of EPS content to support the statement that EPS content “dramatically increased”.
• Line 235 - 236: What are the total Carbon and total Nitrogen values for each media? I assume these values would be used to calculate the ratio. This would help readers to assess whether MVBM has excess carbon.

Effect of culture medium on biochemical composition of B. pseudomallei EPS
• BHI and LB are complex medias – is it possible that residual complex polysaccharides, proteins, and other macromolecules derived from media are present in the EPS?
• Was the total protein content of each media analysed? Does this reflect the total protein content of the EPS?

FTIR spectroscopy analysis of EPS
• Do you have comparative FTIR spectra for each media without bacteria present?
• What controls were used?
• How do these spectra compare to other published data for bacterial biofilms?
• EPS should be a complex mixture of extracellular DNA, polysaccharides, and proteins. What do the FTIR spectra for these “pure” molecules look like?
• Please provide any additional spectral data in the supplementary materials.

Validity of the findings

With regard to "Validity of the findings" issues have been identified with validity of experimental interventions, and use of correlation to imply causation.


Line 238: I question the utility of the post-hoc linear regression analysis.

• If the Carbon to Nitrogen ratio in the supernatant of MVBM culture is changing from Day 2 to Day 6, this may represent a depletion in nitrogen from the media, instead of a secretion of polysaccahrides.
• The change in C/N ratio could also be caused by uncontrolled cell lysis, and/or release of storage polysaccharides (PHB) into the media.
• As the C/N ratio changes over time - the correlation in this graph may merely reflect that the percentage of EPS is increasing along with biofilm maturity. The C/N ratio cannot be meaningfully separated from the time. It would be more accurate to culture bacteria in media with a defined C/N ratio over a set period of time, then measure the EPS percentages. A similar experiment with controlled C/N ratios was cited by the authors in Cui et al., (2017).


Line 274: There are issues with the use of SEM to examine the extracted EPS powder.

• It is difficult to understand what architecture would be remaining after this process.
• The physical arrangement of the residual “EPS” is likely instead representative of entirely separate physical and chemical processes that occur during preparation.
• The considerable issues associated with interpreting EM images of EPS are investigated by Dohnalkova et al., (2010). (https://pmc.ncbi.nlm.nih.gov/articles/PMC3067245/)
• Are the authors satisfied that the information gained by their current SEM protocol is useful?
• The authors acknowledge only in the conclusion that "the extensive processing steps involved in sample preparation may have altered these structures, meaning they might not accurately reflect their native form". Please state this earlier in the discussion.

Additional comments

The authors have significant revisions to make before publication of this manuscript.

The most important changes to be made are:

1. Include all raw data for Total Organic Carbon and Total Kjeldahl Nitrogen for each replicate within the supplementary information. Then the C/N ratio can be accurately interpreted.

2. Provide more methodological details for the experiments performed. In its present form, I have serious concerns as to whether other investigators could replicate this study from the information provided. If the source of a protocol cannot be widely accessed, more detail is needed.

3. Fix referencing errors. Currently, several statements within this study are supported by citations that are either incorrect, or of very poor quality. This decreases trust in the findings and interpretation.

If these issues are satisfactorily addressed, the manuscript will likely be of interest to others in the field.

---

## Round 0.2 · Minor Revisions

· Academic Editor

Minor Revisions

Congratulations on having improved your manuscript. Reviewer 1 has recommended a few additional changes which I agree with, so kindly check those remarks and submit a modified version at your earliest convenience.

Reviewer 1 ·

Basic reporting

The authors made a significant improvement on the manuscript. Overall, the manuscript has a good flow and easier to read and understand.

Experimental design

-

Validity of the findings

1)The paragraph line 418-429 showed contrasting evidence of EPS vs biofilm biomass compared to the authors' study. The authors may describe/elaborate on this.

2)The authors observed that the C/N ratio in nutrient media affect biofilm and EPS formation. Which component plays a more important role, biofilm or EPS, in promoting survival and persistence in BP?

3) What would the authors suggest for nutrient media to be applied for future study involving biofilm investigation?

3)Does the nutrient media affecting the growth of BP ST-39 in this study?

Additional comments

This sentence is confusing.
Line 499 The identify associative trends of C/N association between culture media on EPS reported are correlative in nature and may not be interpreted as evidence of causation.

·

Basic reporting

With regard to basic reporting, I commend the authors for their significant efforts to improve the manuscript. In particular I thank the authors for correcting their references, and providing their readers with the raw data.
I have no further suggestions to make as the basic reporting now meets the standards of PeerJ.

Experimental design

With regard to experimental design, I appreciate the significant edits made by the authors.
Especially important is their inclusion of the full Volatile suspended solids, TOC, and Kjeldahl methods used. The increased level of detail in the biofilm formation assay is also important, as these types of assays are very often extremely difficult to replicate between laboratories.
The formal publication of these method will form a valuable resource for their peers who may wish to utilize similar methods. All methods now have sufficient detail and information for replication.

Validity of the findings

With regard to validity of findings, I thank the authors for their constructive application of the review comments. Associative data is still of significant value to the scientific literature, and is worth publishing. Having addressed the limitations fully in the conclusions, I am satisfied of the benefit to literature and the validity of the findings.

Additional comments

The authors have undergone a vigorous and substantial review process, they are to be commended for their efforts to provide raw data and highly detailed methods. I am satisfied that the revised publication will be well-received by their field.

---

## Round 0.3 · accepted · Accept

· Academic Editor

Accept

I would like to congratulate you for having successfully responded to remarks raised during peer-review of your work. Hopefully, you found those constructive and helpful. I invite you to consider PeerJ for your next submission.